# Measuring Customer Reservation Price for Maintenance, Repair and Operations of the Metro Public Transport System in Qatar

**Jaime Larumbe**

Independent Researcher, 48013 Bilbao, Spain; larumbe.jaime@gmail.com

**Abstract:** Getting to know the price that users assign to maintenance, repair and operations (MRO) has arisen as an essential consideration in gathering financial sustainability for metro public transport systems. The current research reveals customer reservation price for MRO in the main metro stations in Qatar. The purpose of the present work is to assess the willingness to pay for MRO services in eight metro stations in Doha in order to have a better understanding of user preferences. Qualitative research was carried out employing primary and secondary source of information. Primary data was collected by means of a mixture of data accumulation approaches: key informant meetings and focus-group conversations. Secondary data was collected from the account books, contracts, recordings of trans-actions, statements of work and activity reports given by the local rail committees. A stated preference investigation was applied through open text format questions to more than 1000 customers, and a Poisson regression model was used to evaluate the considerations affecting every higher value. Outputs reveal normal customer reservation prices per month and per train journey. The results also indicate a significant willingness to pay differential among the studied railway stations. The study of the decisive considerations elicits that the degree to which the MRO service can exclude paying consumers, the attending of rail conferences and the possibility of using another rail station are related with the customer reservation price. The outputs of this research are significant for railway public authorities willing to set up reasonable, adequate and realistic fares that support fund competent railway systems in Qatar.

**Keywords:** customer reservation price; financial sustainability; metro fares; metro stations; Qatar





## 1. Introduction

Although transport is an agreed enabler of economic growth, the role of sustainable transport is gaining recognition as an enabler of sustainable development. The Railway system is the backbone of sustainable transport. It makes close links to many goals and supporting targets: energy efficiency, resilient infrastructure and access to sustainable transport, resilience to climate-related hazards, fostering environmentally sound technologies and multi-stakeholder partnerships [1].

Inappropriate financing of railway MRO could turn into an obstacle for the supply of reliable railway services in some countries of the Middle East [2,3]. Although remarkable progress has been made in recent years, increasing railway service to remote locations [4,5], this has not translated into reliable service delivery [6,7]. Actually, a 2019 International Railway Agency (UIC) report on the average railway service sector functionality rates indicates that a significant amount of railway systems experience interruptions a few years after their implementation [8].

In this situation, present railway development efforts in the Middle East have focused on giving people access to reliable, adequate and competent railway systems for all [9] under the introduction of the Gulf Railway system [10,11]. Accomplishing these objectives requests a deep comprehension of the conditions that allow the long-lasting performance of railway systems [12,13].

In the last years, the national railway policies of several countries have relied on community-based management as the main model for managing railway transport sys-

tems [14,15]. The community-based management model underlines the financial responsibility of the users to essentially finance the recovery of MRO costs and the long-term replacement capital [16]. In spite of the support of key railway players such as agencies and governments [17–19], recent scrutiny of the long-term performance of railway systems highlights the weaknesses of the community-based management model [20–22] and calls for enhancements in the financial model. Similarly, many studies assessing railway financing challenges propose that a lot of users struggle to set up and maintain a system for gathering user fees and raise the needed funds for sustainable railway MRO cost recovery [23–25].

Under the community-based management model, users are regularly expected to raise amounts which are not acceptable by many members, not affordable by vulnerable users or regularly impracticable as it comes from productive-seasons-based income [26–33].

Several other authors [34–38] have highlighted the necessity of a new model where cost sharing arrangements between users and railway agencies could provide reliable railway service. In addition, international organisations [39–41] recommend an increase in external financial support to supplement user contributions and ensure the financial sustainability of the railway projects. The purpose of the new model is a partial recovery of MRO costs through user fares while transfers from private contributions would bear some major repairs, rehabilitation and replacements.

Regarding this, the UIC railway sector evaluation report highlighted the urgency for a better comprehension of user preferences in the form of customer reservation price. The report finds that an enhanced understanding of users' willingness to pay will support railway agents in establishing correct and affordable fares, which translates into sustainable MRO cost recovery [42–44].

Considering this, the purpose of the present work is to assess the willingness to pay for MRO services in eight metro stations in Doha in order to have a better understanding of user preferences. To the extent of the authors' knowledge, there is no reported study on users' demand for improved railway services in Qatar. The railway agencies in charge of the infrastructures in the city of Qatar for the last years, Qatar Rail and RKH Qitarat, grew concerned about the acute financial challenges for reliable railway service provision. These railway agencies try to set new fares with the users that can be reasonable, adequate and realistic. For this goal, it is necessary to gain a thorough understanding of the users' willingness to pay for MRO services. This is the purpose of the present work.

The paper is structured as follows. The next section presents the materials and methods of the investigation. After that, the results of the research are described. Finally, the paper concludes with a discussion of key findings and a conclusion.

## 2. Materials and Methods

### 2.1. Research Area

Qatar is a country placed in Western Asia, covering the tiny Qatar Peninsula on the northeastern shoreline of the Arabian Peninsula [45,46]. Its only land frontier is with adjacent Gulf Cooperation Council (GCC) monarchy Saudi Arabia to the south, with the rest of its area included in the Persian Gulf.

At the beginning of 2017, Qatar had 2.6 million inhabitants: 313,000 Qatari citizens and 2.3 million expatriates [47]. Islam is the official religion of Qatar [48]. In terms of incomes, the country has the third highest GDP per capita in the world [49], and the sixth highest GNI per capita [50]. Qatar is identified by the UN as a country of very high human development, having the third highest HDI in the Arab world following the United Arab Emirates and Saudi Arabia [51]. Qatar is a World Bank high-income economy, supported by the world's third-biggest natural gas reserve and oil reserve [52]. Qatar is the world's biggest issuer of greenhouse gasses per capita [53].

In comparison with its size, Qatar has a disproportionate weight in the world and has been classified as a middle power [54,55]. The 2022 FIFA World Cup will take place in Qatar, making it the first Muslim and Arab country to welcome the championship [55]. The 2030 Asian Games will be also held in Qatar [56]. With a quick-expanding population

and important economic rise over the last years, a reliable and widespread transportation network is becoming progressively needed within Qatar. So far, the government, the primary transport developer, has done well in regard to catching up with the demand for new transportation possibilities.

In 2008, the Public Works Authority (Ashghal), one of the organizations that supervises infrastructure development, underwent an important reorganisation in order to improve and update the authority in preparation for bigger project expansions across all segments in the short-term. Ashghal works together with the Urban Planning and Development Authority (UPDA), the organization that created the transportation master plan, founded in March 2006 and running until 2025. See Figure 1.

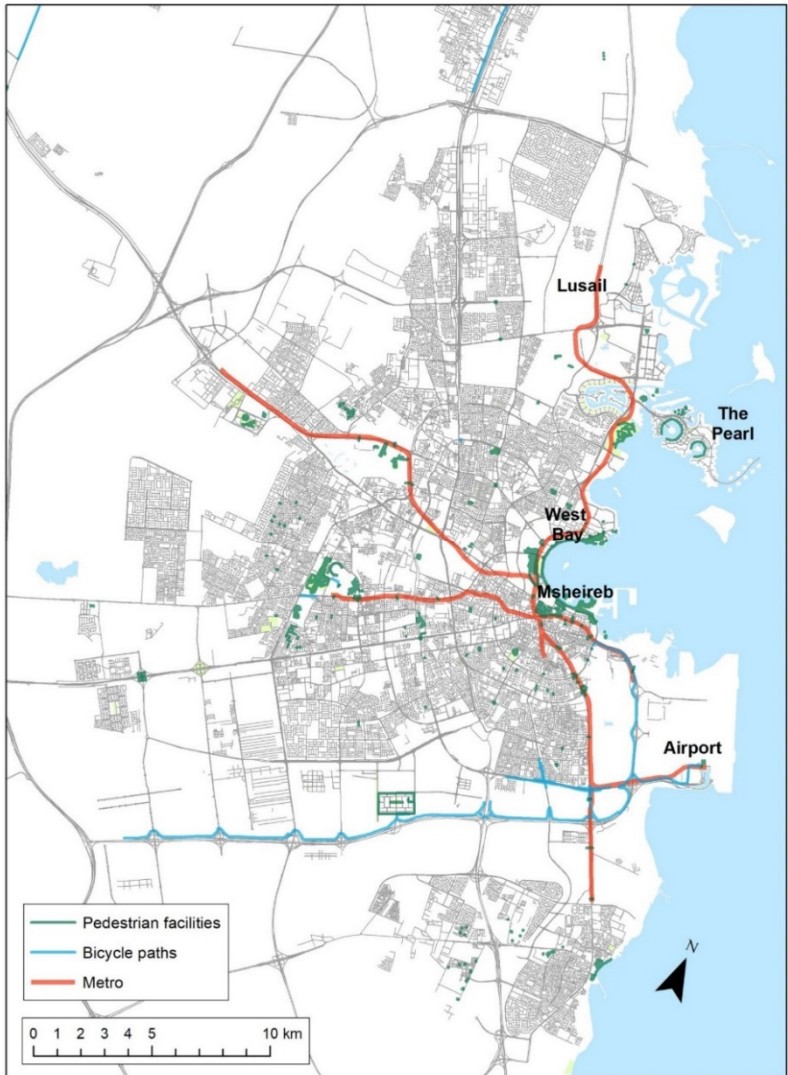

**Figure 1.** Map of the study area.

Mass transit possibilities, such as a Doha metro, light rail system and more extensive bus networks, are also under development to reduce road congestion. In addition, the system is being significantly increased and could eventually form an integral part of a GCC-wide network joining all the Arab states of the Persian Gulf. The airport, too, is increasing capacity to keep up with rising visitor numbers.

Important characteristics of the metro stations are summarised in Table 1. Regarding lines, it is worth saying that whereas only 1 line in West Bay served 1610 thousand users in 2020, in Msheireb, 3 lines served up to 3228 users, and in Al Bidda, 2 lines served up to 2663 users.

**Table 1.** Characteristics of the train station and the sample [57].

| Station 2020 | Lines | Number of Lines | Users per Month in 2020 (Thousand) |
| --- | --- | --- | --- |
| Msheireb | Red, gold, green | 3 | 3228 |
| Souq. Waqif | Gold | 1 | 908 |
| Lusail | Red | 1 | 361 |
| Hamad Intl. Airport | Red | 1 | 1744 |
| West Bay | Red | 1 | 1610 |
| Corniche | Red | 1 | 729 |
| Hamad Hospital | Green | 1 | 256 |
| Al Bidda | Red, green | 2 | 2663 |

*2.2. Data Analysis Framework*

In data analytics, the framework allows you to move through data analysis in an organized way.

During the last decade, a number of studies eliciting customer reservation price for railway services have emerged [58–60]. This interest responds to a historical lack of specific information on community demand for improved railway services in developing countries. In order to implement reliable services, it is necessary to gain a better understanding of the economic value assigned by the users to the improved services [2,44,61]. It is equally crucial to determine the factors conditioning that value.

In this paper, the concept of customer reservation price refers to the stated price that an individual would accept to pay for avoiding the loss or the diminution of a non-market service [62,63].

The empirical literature on users' preferences for railway services indicates a great variety of approaches and analytical techniques for measuring customer reservation price. The main two categories are the Revealed Preference and Stated Preference. In the Revealed Preference approach, the authors perform experiments or simulate price-response data [64]. In the Stated Preference approach, authors employ survey techniques based on hypothetical choices for estimating customer reservation price [65]. Within the Stated Preference option, we can differentiate Direct and Indirect Surveys [66]. To perform Indirect Surveys, a ranking procedure is applied [67]. To perform Direct Surveys, 3 elements are involved: a detailed description of the service offered, a description of how the service would be provided and a method for eliciting preferences for the services [68]. The direct approach is referred to as Contingent Valuation Method (CVM) and despite controversies [69,70], it has become one of the most widely used techniques for valuing non-market goods [71,72]. The present research investigates users' preferences for improved railway services through a CVM survey.

*2.3. Data Gathering Strategy*

2.3.1. Qualitative Research

Qualitative research was carried out employing primary and secondary source of information. It pursued assessing the degree to which the MRO service can exclude paying consumers and reaching an improved comprehension of the variables to incorporate for the quantitative research. Primary data was collected by means of a mixture of data accumulation approaches: key informant meetings and focus-group conversations. Thus, 10 key informant meetings took place at the headquarters of Qatar Rail (QR) applying a structured procedure. The key informants were chosen to compare the stakeholders' views between local authorities accountable for inspection of the metro stations, railway technician people in charge of the control and maintenance of the railway systems, social public authorities responsible for the report of the accounts of the railway services and railway committee board members. The data collected had to do with service supply, use of rail transport, organization and control of railway resources.

Focus-group conversations also took place at the headquarters of QR to collect qualitative data. The conversations were held at an early phase of the field survey. The aim

was to adjust the questions contained in the questionnaire and to describe the criteria to calculate the degree to which the MRO service can exclude paying consumers. Participating people were chosen between railway system beneficiaries, railway committee members and experts, making a distinction between classifications by gender.

Secondary information was gathered from the following:

- Books of accounts reviewing the cash received and expended, sales and purchases of goods and all assets and liabilities of the rail agencies;
- Contracts for zone works and engineering materials, writing down all works of repairs and maintenance plus the conveyance of materials, e.g., bricks, lime, sand and so on, which are likely to be required in an area during the year 2020 for the rail agencies;
- Recordings of internal transactions, extracting information about interdepartmental exchanges of assets, plus external transactions, collecting the exchanges between the rail agencies and suppliers;
- Statement of work paying attention to detailed requirements and pricing, with standard regulatory and governance terms and conditions.

### 2.3.2. Quantitative Research

The eliciting protocol chosen for this research is an open text format survey in which participants are introduced with the benefits of possible services and are queried for the highest prices to pay for them [72]. This procedure has been given priority over the yes/no selection bids to elude final customer reservation price answers to be affected by the kick-off bid and to reduce hypothetical mistakes [67,73]. The CVM investigation was applied by means of personal meetings relying on the organized quiz, which had been tested for internal validity and reliability within a limited sampling of people. The objective of the test was to assure a crystal-clear comprehension of the quiz by local participants. Previous test outputs were applied to improve the quiz accordingly.

The investigation was applied across 8 metro stations over a duration of 1 month in December 2019. The meetings were held by 11 interviewers. They were instructed to make participants feel confident. Prior to taking part, every participant provided oral consent to the involvement in the unknown identity investigation consciously, on the comprehension that the interviewee could quit anytime.

The last edition of the questionnaire consists of 4 parts. In chapter 1, partakers were enquired about their use of the metro. In chapter 2, the CVM is included. Chapter 3 incorporated the decisive considerations of customer reservation price extracted from the qualitative study. In chapter 4, sociodemographic features were handled. The survey was created in English and after that translated into Arabic to be better followed by the partakers.

Significant consideration has been taken in the composition to validity and reliability. While validity refers to the relation among what is supposed to be evaluated and what is really evaluated, reliability refers to the evaluation replicability [67]. In this research, strategic and hypothetical areas of influence were taken into account [74] for our CVM-based predictions to be valid and reliable.

### 2.4. Variables

Qualitative study gives us decisive considerations of customer reservation price over which the MRO service agency predicts remarkable supervision. Consequently, such considerations are to be used as unbiased variables for the quantitative research. Subsequently, 6 variables were chosen: excludability, reliability, conference, duration, options and expense.

In addition, information collected from key informant meetings, focus-group conversations and technical evaluation was also used to calculate how excludable a metro station is. Based on the theory of public goods [75], the extent to which the MRO can exclude paying users depicts the capacity of a metro station to exclude non-paying users.

Higher degrees of excludability are applied with the aim of being more efficient in terms of energy consumption, space occupancy and number of users transported avoiding waiting lines.

Qatar Rail is empowered to manage its particular norms and decrees. During COVID-19, they have imposed stricter regulation, and therefore, they have reduced the amount of users. They have become more exclusive. Table 2 defines the variables, motivation to be chosen and the level to be classified in.

**Table 2.** Independent variables description, motivation and level.

| Variable | Description | Motivation | Level |
|---|---|---|---|
| Excludability | Improved rail system user reservation price | Citizens living in a more exclusive neighbourhood present higher user reservation price [1–3] | 0 to 13 |
| Reliability | Level of reliability of the improved metro system operating safe transport | Reliability of rail systems is related to a financial cost that citizens manage to use [6–8] | 0: non-operational 10/4: suspended every second day 20/4: suspended once per month 30/4: suspended once every 6 months 10: continuous provision |
| Conference | Regular attendance to rail conferences arranged by the government | Users who usually attend rail conferences are more conscious of the budgetary necessities of the rail system [4] | 0: No 10: Yes |
| Duration | Duration in taking the train at another train station. | User reservation price rises if duration to take the train is higher, presuming that opportunity costs will be improved [9,10] | 0: less than 15 min 10/3: between 15 and 30 min 20/3: between 31 and 60 min 10: over 60 min |
| Options | Distance from optional metro station to users | Citizens request of improved metro transport service depends on existing options [5,62] | 0: options placed close to users 10: options placed distant from users |
| Expense | Level of user expense per month | Citizens who reveal higher monthly expenses have more resources to pay their rail tariffs [1–3] | 0: less than USD 1000 * 10/3: between USD 1000 and USD 3000 20/3: between USD 3100 and USD 6000 10: more than USD 6000 * |

* Note: Exchange rates on 13 August 2021: USD 1 = QAR 3.64.

In addition, data gathered from key informant interviews, focus group discussions and technical assessment was also employed to measure the level of excludability for each station.

Based on the theory of public goods [63], the level of excludability indicates the ability of a station to exclude non-paying users. Higher levels of excludability are implemented with the intention of preventing queues, wear of the stations and potential rationing of the rail service.

In Qatar, the railway committees are able to administer their own rules and regulations. Some groups prefer to impose tighter regulations, and thus, they reduce the number of users. They become more exclusive. In order to measure the levels of excludability, four categories of excludability were analyzed in this study: operational, social, physical and managerial [64].

Operational excludability assesses the establishment of railway fee prices, payment places and dates and the functioning of a cashier role.

Social excludability weighs the establishment of schedules for railway usage and the organization of periodical meetings and general assemblies.

Physical excludability evaluates the use of locks, fences and railway point attendants employed to control the access to services.

Managerial factors assess the use of delinquency lists, user directory, accounting books and the adoption of fines and/or exclusion procedures.

By quantifying these factors of excludability set in place at the public stations, the stations were classified into three levels of excludability: Big (B: 10–13), Medium (M: 4–9) and Small (S: 0–3) (Table 3).

**Table 3.** Considerations of excludability quantified for every metro station.

| Station | Physical | Managerial | Operational | Social | Weight | Excludability |
|---|---|---|---|---|---|---|
| Msheireb | 2 | 1 | 2 | 2 | 7 | M |
| Souq Waqif | 2 | 2 | 2 | 2 | 8 | M |
| Lusail | 2 | 2 | 2 | 2 | 8 | M |
| Hamad Intl. Airport | 1 | 0 | 1 | 0 | 2 | S |
| West Bay | 3 | 4 | 3 | 3 | 13 | B |
| Corniche | 0 | 0 | 1 | 1 | 2 | S |
| Hamad Hospital | 2 | 1 | 2 | 2 | 7 | M |
| Al Bidda | 2 | 0 | 2 | 2 | 6 | M |

Note: Error for 95% confidence level.

### 2.5. Sampling Frame

The focus of this research is limited to Doha. Sampling population and the sampling technique were concretely planned to fulfil the investigation objectives (see Table 4).

**Table 4.** Features of the population and the sampling.

| Station | Users per Month in 2020 (Thousand) | Participants | % Error | % Participation |
|---|---|---|---|---|
| Msheireb | 3228 | 140 | 8.1 | 4.34 |
| Souq. Waqif | 908 | 144 | 7.5 | 15.86 |
| Lusail | 361 | 142 | 6.41 | 39.34 |
| Hamad Intl. Airport | 1744 | 140 | 7.95 | 8.03 |
| West Bay | 1610 | 143 | 7.69 | 8.88 |
| Corniche | 729 | 107 | 8.76 | 14.68 |
| Hamad Hospital | 256 | 149 | 5.2 | 58.2 |
| Al Bidda | 2663 | 144 | 7.94 | 5.41 |
| Total | 11,499 | 1109 | 2.8 | 9.64 |

The investigation was conducted combining 2 different sample approaches. A purposive sample technique was carried out for the 8 metro stations, while a systematic sampling technic was undertaken to investigate the beneficiary users.

Every station was subdivided into areas of analogous population density. Interviewers undertook a set amount of interviews for every area. The interviewers were positioned at every metro station gate. From their locations, they selected users of the rail service at random to survey.

### 2.6. Statistic Investigation

Descriptive statistics investigated the mean values of customer reservation price and contrasted the weights of considerations determining customer reservation price for every metro station involved. The relation among the considerations and the dependent variables $CRP_m$ (Customer reservation price per month) and $CRP_j$ (Customer reservation price per train journey) were previously checked through a correlation study. At the end, a linear relation was hypothesized among a log transformed customer reservation price and the predictor variables. The predictive patterns comply with the following algorithm:

$$\log (Y) = \beta_0 + \beta_1 \cdot X_1 + \beta_2 \cdot X_2 + ... + \beta_n \cdot X_n + \varepsilon \tag{1}$$

where Y is the dependent variable, $X_i$ is the predictor variables and $\beta_0$ is the mean of the answer if every predictor is 0. Each factor calculates the weight of the independent variable in the pattern, and $\varepsilon$ is the error of the prediction. The candidate variables have been

summed up through a forward gradual method. The suitability of the pattern is measured calculating the described variance $R_2$, the F test, the regression error and the significance of the factors [76,77]. Data analysis was developed by means of IBM SPSS, Version 26.0. (IBM Corp., Armonk, NY, USA).

## 3. Outputs

Users' features are introduced in Table 5. Almost 73% of the participants (N = 1109) in the investigation were female, and approximately two thirds were older than 25 and younger than 60 years old. Between them, nearly 34% had not any official education, next to 50% had finished primary school and about 22% had finished secondary school or above. Regarding employment, close to two-thirds worked for the oil sector, almost 24% were employed in commerce and just 13% worked for other areas. Above 47% of this sampling revealed having five to eight members in the family.

**Table 5.** Sociodemographic features of the research sampling.

| Gender | | | Age | | | Employment | | | | | |
|---|---|---|---|---|---|---|---|---|---|---|---|
| | | | | | | Commerce | | Oil Sector | | Others | |
| **Male** | | **Female** | **Over 60** | **26–60** | **Under 25** | **Male** | **Female** | **Male** | **Female** | **Male** | **Female** |
| 298 | | 801 | 96 | 754 | 243 | 209.6 | 52.4 | 591.6 | 104.4 | 155.2 | 38.8 |

| Education | | | | | | Family members | | | |
|---|---|---|---|---|---|---|---|---|---|
| Secondary | | Primary | | Unofficial | | +12 | 9–12 | 5–8 | 2–4 |
| Male | Female | Male | Female | Male | Female | | | | |
| 142.8 | 95.2 | 324.35 | 174.65 | 241.15 | 129.85 | 95 | 244 | 522 | 245 |

A synopsis of descriptive statistics for the group of variables used in the research is explained on Table 6. The mean $CRP_j$ has been evaluated at QAR 36 and $CRP_m$ at QAR 327. These prices indicate the average estimated customer reservation price for MRO of metro services in Qatar. An important differential customer reservation price can be noticed between metro stations. The $CRP_m$ ranges from QAR 730 in West Bay to QAR 173 in Hamad Intl. Airport and 193 in Al Bidda. The $CRP_j$ ranges from QAR 71 in Hamad Hospital to less than QAR 20 for Msheireb, Hamad Intl. Airport and Corniche. Furthermore, important divergences were recognized ($\alpha = 10^{-2}$) for the existing variables. The mean value for expenses in Hamad Intl. Airport is assessed at 0.41, almost 10 times lower than the degree of expenses in other stations such as Lusail and West Bay. The mean value for Options ranges from 7.04 in Hamad Hospital to 0.98 Souq. Waqif. The average degree of reliability of the metro service is rated at 6.39 in Lusail and at 0.93 Hamad Hospital. The population of Qatar is very diverse as we can observe from the difference of the mean values.

**Table 6.** Mean values of $CRP_j$, $CRP_m$ and predictor variables.

| Station | N | Mean (SD) | | | | | | |
|---|---|---|---|---|---|---|---|---|
| | | Options | Reliablility | Duration | Expense | Conference | $CRP_m$ | $CRP_j$ |
| Msheireb | 140 | 5.64 (4.98) | 5.02 (2.80) | 3.36 (2.63) | 4.22 (2.05) | 2.88 (4.54) | 268.71 (262.89) | 16.87 (41.09) |
| Souq. Waqif | 144 | 5.55 (4.99) | 4.62 (2.06) | 2.73 (2.70) | 4.52 (2.00) | 4.48 (4.99) | 287.80 (156.15) | 25.77 (45.38) |
| Lusail | 142 | 6.40 (4.82) | 6.39 (2.39) | 4.76 (2.91) | 5.64 (2.44) | 7.86 (4.12) | 261.76 (175.50) | 60.42 (64.43) |
| Hamad Intl. Airport | 140 | 0.98 (2.99) | 0.93 (2.53) | 5.24 (3.73) | 0.41 (1.17) | 3.77 (4.86) | 173.93 (246.44) | 15.32 (52.20) |
| West Bay | 143 | 6.67 (4.74) | 5.86 (2.75) | 2.77 (2.62) | 5.41 (2.50) | 3.66 (4.83) | 730.28 (1052.52) | 51.05 (51.95) |
| Corniche | 107 | 3.94 (4.91) | 4.55 (3.48) | 4.62 (2.89) | 3.52 (2.99) | 4.52 (5.00) | 234.86 (226.62) | 17.56 (28.53) |
| Hamad Hospital | 149 | 7.04 (4.58) | 3.96 (2.64) | 3.49 (2.39) | 4.27 (3.22) | 6.33 (4.84) | 439.38 (350.52) | 71.14 (125.69) |
| Al Bidda | 144 | 4.41 (4.99) | 5.40 (3.75) | 3.96 (3.37) | 2.81 (2.85) | 4.76 (5.01) | 193.40 (285.79) | 27.78 (60.70) |

In order to evaluate the intensity of the relations among willingness to pay and its conditioning factors, a correlation analysis in three steps was developed. First, it is noted



that excludability is the variable exerting the biggest effect over log $(CRP_j)$ and log $(CRP_m)$: significant Pearson correlation indicators for $\alpha = 10^{-2}$ of $39.8 \times 10^{-2}$ and $39.3 \times 10^{-2}$ (Table 7). In addition, all the variables investigated have a positive and important correlation with log$(CRP_j)$ and log$(CRP_m)$. The correlation indices evidence that expenses stands out with values of 0.248 and 0.300 for both results, as can be observed in Table 7.

**Table 7.** Correlation indicators for log $(CRP_j)$, log $(CRP_m)$ and predictors.

| Predictors | Log $(CRP_m)$ | Log $(CRP_j)$ |
|---|---|---|
| Reliability | 0.168 ** | 0.216 ** |
| Expense | 0.300 ** | 0.248 ** |
| Conference | 0.067 * | 0.158 ** |
| Duration | 0.089 ** | 0.130 ** |
| Options | 0.119 ** | 0.248 ** |
| Excludability | 0.393 ** | 0.398 ** |

Note: * $p < 0.05$ and ** $p < 0.01$ in Pearson correlation tests (two-tailed); cell entries are standardized coefficients.

Having recognized important discrepancies between metro stations for every predictor, in a second stage, a correlational research categorized by metro stations was undertaken. Consequently, a study categorized by degrees to which the MRO service can exclude paying consumers was also conducted. The outputs show that in West Bay, a negative high correlation among log$(CRP_j)$ and conference attendance was found (Table 8). In addition, a negative correlation is depicted among log$(CRP_m)$ and conference connected to small degree of excludability, obtaining a 0.472 factor in Souq Waqif. However, the correlation between conference and log$(CRP_j)$ turns positive for the modest degree of excludability, being relevant in Hamad Hospital, Msheireb, Al Bidda and Souq Waqif.

**Table 8.** Correlation study for log$(CRP_j)$, log$(CRP_m)$ categorised by metro stations and degree of excludability.

| Station | Log $(CRP_j)$ | | | | | Log $(CRP_m)$ | | | | |
|---|---|---|---|---|---|---|---|---|---|---|
| | Reliability | Expense | Conf. | Duration | Options | Reliability | Expense | Conf. | Duration | Options |
| Msheireb | −0.094 | 0.278 ** | 0.269 ** | 0.141 | 0.088 | 0.037 | 0.140 | 0.144 | 0.144 | 0.028 |
| Souq. Waq. | 0.249 ** | 0.163 | 0.249 ** | −0.148 | −0.079 | −0.135 | 0.072 | −0.172 | −0.194 | 0.036 |
| Lusail | 0.138 | 0.024 | 0.097 | 0.028 | 0.111 | −0.028 | 0.013 | −0.093 | 0.011 | −0.185 |
| Hamad I. A. | 0.018 | 0.075 | −0.007 | 0.191 * | −0.094 | 0.040 | 0.163 | 0.472 ** | 0.611 ** | 0.143 |
| West Bay | −0.053 | −0.117 | 0.341 ** | 0.021 | −0.081 | −0.049 | −0.017 | 0.054 | −0.059 | 0.080 |
| Corniche | 0.032 | −0.131 | 0.018 | −0.053 | −0.188 | 0.050 | −0.030 | 0.142 | 0.039 | 0.280 ** |
| Hamad H. | −0.053 | −0.113 | 0.239 ** | −0.165 * | 0.164 | 0.035 | 0.054 | 0.077 | 0.087 | 0.022 |
| Al Bidda | −0.079 | −0.193 * | 0.296 ** | −0.073 | 0.133 | −0.161 | 0.197 * | −0.061 | 0.255 ** | 0.392 ** |
| **Level of Interch.** | Log $(CRP_j)$ | | | | | Log $(CRP_m)$ | | | | |
| | Reliability | Expense | Conf. | Duration | Options | Reliability | Expense | Conf. | Duration | Options |
| Total small | 0.137 * | 0.094 | 0.022 | 0.076 | −0.078 | 0.226 ** | 0.271 ** | 0.270 ** | 0.424 ** | 0.167 * |
| Total med. | −0.046 | 0.146 ** | 0.359 ** | 0.029 | 0.135 ** | 0.110 ** | 0.166 ** | 0.018 | 0.066 | 0.037 ** |
| Total big | −0.053 | −0.117 | 0.341 ** | 0.021 | −0.081 | −0.049 | −0.017 | 0.054 | −0.059 | 0.080 |

Note: * $p < 0.05$ and ** $p < 0.01$ in Pearson correlation tests (two-tailed); cell entries are standardized coefficients.

The variable reliability has a negative correlation with log $(CRP_m)$ for the modest degrees to which the MRO service can exclude paying consumers but depicts a positive relation with log $(CRP_j)$ and log $(CRP_m)$ to a modest degree. The correlation of expense with log $(CRP_m)$ is positive and important for the modest and big degrees, to which the MRO service can be paid by customers, whereas the correlation with log $(CRP_j)$ is positive and relevant just for the modest degree. Duration has a relevant correlation with log $(CRP_m)$ for the small degree to which the MRO service can be paid by customers, underlining the intensity of the relation for the case of Hamad Intl. Airport (0.611). The outputs also pin-point a positive correlation for options with log $(CRP_j)$ for the modest excludability and with log $(CRP_m)$ for the small degree.

A multi-regression research has been applied to calculate the relation among customer reservation price and the dependent variables by means of a predictive model. In an initial evaluation, the collinearity of the independent variables was studied. Concretely, a main component regression was developed for this objective. Thus, the variables of excludability, duration and conference were obtained. These variables justified 63.3% of the total variance.

The remaining variables, reliability, options and expenses, are also strongly associated with log ($CRP_j$) and log ($CRP_m$). In spite of that, these predictors are concurrently correlated for $\alpha = 10^{-2}$ with excludability, pinpointing correlation indicators of 0.331 for reliability, 0.252 for options and 0.428 for expenses. Thus, they have been ignored from the patterns [78].

A linear multi-regression pattern with logarithmic transformations (Poisson regression model) was chosen to describe the variables $CRP_j$ and $CRP_m$. A progressive stepwise approach has been applied. For both, the regression studies are significant for $\alpha = 10^{-2}$ with F values of 93.222 and 130.093. The values of $R^2$ are reasonable (0.193 and 0.205) given the complexity of the research, the amount of independent variables in every regression (2 and 3), the significance of the factors and the number of cases employed in the study [78].

As foreseen from the correlation study, excludability exercises the strictest impact on the independent variables. In the first stage, this variable exercises a positive impact on log($CRP_j$), describing 15.6% of its variance with a factor value of 0.098 (Table 9). In the second and third stages, the involvement of the variables conference and duration enhance the pattern, increasing the described variance from 15.6% to 20.5%, but the F value drops significantly from 201.041 to 93.222. In this final pattern, the factors are 0.101 for excludability, 0.035 for conference and 0.018 for duration.

**Table 9.** Predictive patterns for log ($CRP_j$) and log ($CRP_m$).

| | **Log ($CRP_j$)** | | | | | | **Log ($CRP_m$).** | | | | | | |
|---|---|---|---|---|---|---|---|---|---|---|---|---|---|
| **Variable** | **F** | **R²** | **SE** | **Model Sig** | **Variable Sig** | **β** | **Variable** | **F** | **R²** | **SE** | **Model Sig** | **Variable Sig** | **β** |
| Constant excludability | 209.43 | 0.162 | 0.523 | 0.000 | 0.000 0.000 | 1.801 0.070 | Constant excludability | 201.04 | 0.156 | 0.742 | 0.000 | 0.000 0.000 | 0.314 0.098 |
| Constant excludability time | 130.09 | 0.193 | 0.513 | 0.000 | 0.000 0.000 0.000 | 1.625 0.077 0.034 | Constant excludability conference | 136.07 | 0.200 | 0.722 | 0.000 | 0.004 0.000 0.000 | 0.156 0.097 0.034 |
| | | | | | | | Constant excludability conference duration | 93.22 | 0.205 | 0.720 | 0.000 | 0.420 0.000 0.000 0.013 | 0.055 0.101 0.035 0.018 |

Modelling log ($CRP_m$), the variable excludability justifies 16.3% of the variance. The variable duration is involved in the second stage. It increases the $R^2$ from 0.162 to 0.193, but it reduces the F value from 209.425 to 130.093. In spite of that, it should be underlined that the significance degree of the variables is 0.000. The variable conference is not involved in the pattern because its involvement is not relevant in the third step. The factors β are positive in the two stages. The value of β for the variable excludability in the first stage is 0.070, and its impact increases to 0.077 in the second step. The value of β for the variable duration is 0.034. Consequently, to these outputs, the predictive patterns for log($CRP_j$) and log($CRP_m$) fulfil the following algorithms:

$$\log(CRP_j) = 0.055 + 0.101 \text{ excludability} + 0.035 \text{ conference} + 0.018 \text{ duration} \quad (2)$$

$$\log(CRP_m) = 1.625 + 0.077 \text{ excludability} + 0.034 \text{ duration} \quad (3)$$

## 4. Discussion

The financial sustainability of the metro MRO depends mostly on continuous users' revenue collection. Present railway models support partial cost recovery for MRO by means of financing strategy shared by the local government and users. The creation of correct metro fares is important to obtain continuous user payment in the long-term. To achieve this, it is key to assess users' demands for the railway service provided and to set up the price that customers place on such services.

This research reveals customer reservation price for MRO across eight metro stations. The outputs of the investigation evidence three key insights. First, the average economic value that users in Qatar attribute to frequent MRO of the enhanced metro systems has been rated at QAR 327 per month. The outputs disclose a relevant $CRP_m$ differential between the analyzed stations of up to 420%. Disclosed $CRP_m$ evaluations range from QAR 730 in West Bay to QAR 174 in Hamad Intl. Airport. These estimations depict the user preferences for reasonable service at every metro station. These values enable us to comprehend the degree of payment that metro users from local stations feel capable or willing to bear. This information on the local need for enhanced rail service is of paramount relevance for the regulation authority to set up reasonable metro fares.

A substantially high percentage of beneficiaries from the metro systems requested payable MRO. Above 92% of participants revealed their preference for an enhanced service and showed their aim to pay for MRO. The evidence proposes that revenue collection systems are expected to be sustainable if user payment fee exceeds a 60% threshold [79], disclosing the potential for metro stations in Qatar to achieve stable revenue collection systems. In spite of that, frequently, there is a contradiction between users showing demand for improved systems and their contribution behavior for true expenses [79]. International agencies disclose that, across the Middle East, few of the interventions keep an operational system of railway fares many years after implementation [80].

Furthermore, the whole revenue collected from user payments when implementing the declared preferred monthly payment of QAR 327 would be distant from ensuring full life cycle cost recovery for the metro stations in Doha. Documentary information from the surveyed stations disclose that lately some important repairs have exceeded an expense of QAR 25.106. In spite of that, in the creation of a cost-sharing arrangement that QR and RKH Qitarat is about to implement, railway fares are expected to bear just small repairs and preventative maintenance while each major repair, rehabilitation and replacement should be covered by a revenue from government subsidies. Only if the new railway fares are reasonable and adequate could the revenue collected from users' payment be sustained in the long-term and contribute to the economic sustainability of the railway systems.

On the other hand, the identification of other contribution alternatives could play a key part in setting up fares adequate for all, including the poorest. The average declared customer reservation price per train journey has been estimated at QAR 36, varying from QAR 71 in Hamad-Hospital to QAR 15 in Hamad-Itl. Airport. This alternative involves a higher fee per km of railway (e.g., the total expense in railway for a citizen considering a consumption of one train journey per day would result in QAR 1100 per month). In spite of that, this alternative is more practicable for users in informal economies who have difficulties collecting the quantity needed for a monthly single instalment. The pay per train journey mode offers people with limited financial resources an alternative to make use of the metro service on a daily basis.

Furthermore, outputs disclose three main determinant factors of customer reservation price for MRO in Qatar. The predictive model is mostly driven by the level of excludability to use enhanced railway systems, the involvement of users in railway conferences arranged by QR and RKH Qitarat and the level of personal expenditure per month.

In spite of the proximity and in spite of having been applied by the similar agency in identical organization charts, stations have evolved to new ways of management. The collective nature of management of railway services has translated into user groups choosing new levels of excludability to provide long term performance of the metro stations. Preced-

ing investigations proposed that implementing public goods principles to the collective management of metro services could be an adequate framework of evaluation [81]. In Qatar, more exclusive associations have established tighter financial regulations to generate the required revenue. They have reached this because of affiliation fees, penalties and stronger imposition instruments. Users of systems with exclusive ways of management show higher average customer reservation prices per month than more inclusive associations. The outputs of this research suggest that in Qatar a more exclusive way of management is related to higher user contribution levels.

Similarly, the outputs also imply that users that spend more time taking the metro from an improved station are more expected to admit a bigger customer reservation price for MRO. Key informants said that due to present inefficiencies in MRO, a metro station became non-functional. Users of that metro stations were pushed to walk farther, looking for an enhanced station. Therefore, they have seen their time employed in taking the metro increase considerably. Users who request more time to take the train are willing to pay higher metro fees in order to enhance MRO and to restore functioning metro stations.

Furthermore, taking part in conferences held by the railway committees leads to a higher customer reservation price between participants. Throughout the focus group debates, some participants proposed that rail users who often take part in informative and managerial conferences with the committees become more conscious of the financial challenges addressed by the systems and consequently report stronger preferences for MRO.

In this research, many potential constraints have been depicted. The unique social, cultural and political landscape of Qatar constrains the potential generalization of the findings. The group of interviewers for the investigation was selected from among engineers working at the railway operator. This may have affected rail users' responses in spite of strong ethical policy. To reduce the influence, all enumerators were assigned to survey a metro station new from the station they often work at. Protest votes and yea-sayers were not identified with further questions, which could introduce downward bias into customer reservation price evaluations. Eventually, there are also issues about potential strategic and hypothetical biases associated with the CVM questionnaire. Strategic bias in this investigation emerges because interviewees may voluntarily articulate their replies to affect the result of the research in their own profit [82]. Hypothetical bias appears because there is no real purchase of the service, which generally turns into an overestimation of customer reservation price [82]. Being conscious of these concerns, it is needed to be conservative with the estimations and related repercussions.

## 5. Conclusions

The results of this research are the following:

- Improved understanding of customer reservation price for the railway MRO in Qatar;
- High demand of the MRO service by metro system Qatar user of the metro systems
- The economic value that beneficiaries of the systems attribute to improved services is QAR 327 per month;
- An important $CRP_m$ differential has been noticed between users from different metro stations;
- A significant proportion of users would benefit from a pay per train journey, in spite of the rise in the fare up to QAR 36 per train journey;
- The level of excludability of the railway MRO service, the involvement in rail management conferences and the time employed in taking the metro from an improved metro station are contingent with customer reservation price.

An improved comprehension of local requests for MRO service in Qatar is of paramount importance for railway agencies working in the area. In an attempt to achieve financial sustainability, key public rail authorities will make use of these results to establish metro fares that can be realistic, reasonable and adequate for all.

As a result, this article will have several key economic policy contributions. The railway systems will have funds to meet all the MRO service milestones and financial obligations. They will have the capacity to generate the expected return for the investors. The railway transport will be offered at a price that not only covers expenses but also creates a profit. In a crisis, it will involve meeting financial obligations within have other alternative possibilities to fund railway services. In other words, it will be able to be self-financed. Finally, the railway agencies will have the capacity to obtain revenues in response to a demand in order to sustain MRO processes at a steady or growing rate to produce results and obtain a surplus.

**Funding:** This research received no external or internal funding.

**Institutional Review Board Statement:** The research was conducted according to the guidelines of the Declaration of Helsinki and approved by QR Ethics Committee on 2 October 2020.

**Informed Consent Statement:** Informed consent was obtained from all subjects involved in the research.

**Data Availability Statement:** The data presented in this research are available on request from the corresponding author.

**Acknowledgments:** The authors would like to thank the QR Authorities in the person of his Excellency Abdulla bin Abdulaziz bin Turki Al Subaie for supporting the work reported above. The findings and conclusions expressed in this paper though, represent exclusively those of the author.

**Conflicts of Interest:** The author declares no conflict of interests.

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
