# Peer review of "Measuring Customer Reservation Price for Maintenance, Repair and Operations of the Metro Public Transport System in Qatar"

_sustainability, doi:10.3390/su131911023_

Round 1
Reviewer 1 Report
Dear author,
First of all - congratulations on a very interesting research! The article "Measuring Customer Reservation Price for Maintenance, Repair and Operations of the Metro Public Transport System in Qatar" is well prepared.
Nevertheless, I have some minor comments to your article.
1) Abstract / Introduction
There is no clearly defined research goal - I don't see it in Abstract neither in Introduction. Please put the purpose of the work in the article.
2) Materials and Methods / Outputs
Try to correct the tables 1, 3, 4, 6 and 8.
The "Hamad Intl. Airport" station is divided into two lines" "Hamad Intl." and "Airport". It looks like two stations. The same is true for the "Hamad Hospital" station, for example.
Try to put each station name on one line only.
3) Conclusions
This part is too general. There are no clearly defined results - e.g. bulleted results.
Emphasize the practical value of the article more.
Author Response
- Research goal introduced in the abstract and introduction
- Name of the stations on one line for all the tables
- Practical value of the article emphasised and the defined results bulleted
Reviewer 2 Report
Dear author
I would like to rise the following comments for your manuscript titled (Measuring Customer Reservation Price for Maintenance, Repair and Operations of the Metro Public Transport System in Qatar):
- The literature on the topic needs more resources to cover the topic and why your research is relevant in terms of methodology and planning policy. As well as, there is no structure of the article in the introduction section.
- This article missed the innovation of methodological development.
- The discussion and conclusion sections are not clear, please revise it.
Best regards
Author Response
The literature of the article on the topic has been improved.
The relevance of the research in terms of methodology and planning policy has been highlighted in the introduction, discussion and rest of the rest of the article.
The structure of the article has been included at the end of the introduction.
The discussion and conclusion sections have been clarified.
Reviewer 3 Report
The manuscript entitled Measuring Customer Reservation Price for Maintenance, Repair and Operations of the Metro Public Transport System in Qatar is well written, in particular the section devoted to data and the analyses. Unfortunately, the Introduction and Discussion together with the Conclusion are not sufficient to accept this manuscript for publication in Sustainability. I recommend to the author to go through my general suggestions and improve the paper.
In the Introduction the authors of the manuscript have clearly explained the rationale for the study. However, taking into consideration the scope of the Sustainability journal, for which the issues and challenges connected with sustainable development are crucial, I recommend authors add at least one paragraph which would place this study into the context of sustainable development.
Also, and this I consider as a serious flaw, there is not sufficient Literature review in the manuscript. It is true, the author makes comments in the Introduction and 2nd section, but this is only very little. I recommend, for example, creating a subsection Introduction where the author will present a brief overview of important themes of this study, such as the importance of price in regards to public transport/metro and connected elements this study focuses on.
Lastly, the Discussion section is more of a conclusion. I suggest rewriting those parts. For the Discussion, the author should show the result of this study in comparison with findings of similar studies and place them in the context by providing a commentary. Also, sustainable development should be covered anyhow in the Discussion section. Policy recommendations should appear in the Conclusion as well, as, in the current form, there aren't any.
The Ethic statement section should be at the end of the manuscript in the section where acknowledgement is.
Detailed comments:
Table 1 – a missing source of the data
Table 5 – please, present the male/female proportions not only for the total sample but also according to the categories (education, employment)
Author Response
Paragraph to place the study into the context of the sustainable development included in the introduction.
Literature review of the article improved.
Overview of the importance of price in regards to public transport included in the introduction and discussion.
Discussion and conclusion re-written.
Findings of similar studies placed in the context provided.
Sustainable development covered.
Policy recommendations included.
Ethic statement moved to the end of the article.
Table 1 source of data included.
Table 5 male/female proportions included.
Reviewer 4 Report
This manuscript examines an enhanced comprehension of customer reservation price for maintenance, repair and operations (MRO), using Qatar as a case study. The key theme should have been addressed clearly, as I found that it is sometimes difficult to follow. In addition, the structure and logic could be improved. What are the research questions? The literature review combined with the introduction is fairly descriptive. Given that this is an academic journal paper, there is no need to highlight ‘the report xxx’. The introduction of the case study Qatar is good, but this should always relate to railway MRO, not something else. The qualitative research method is introduced but without having any further detailed analyses in the findings. Therefore, I would suggest only keeping the quantitative part. In terms of the quantitative research method, some of the specific tools are unclear. For instance, the SP investigations are difficult to follow. There are some typos across the entire manuscript, which should have been avoided before submission. Detailed comments are provided below:
Abstract
- In the abstract, page 1, line 10, the statement of what the paper highlights need to be more precise. For instance, to what extent the customer reservation price plays a role in PT systems.
- Page 1, line 11, the sentence ‘Outputs also show…’ is not clear.
- Page 1, this research seems to use both qualitative and quantitative methods. However, after reading the Abstract, it does not show how the qualitative method was applied/worked out.
Introduction
- Pages 1 and 2, I would suggest revising and clarifying the research background, aim, questions, and research gap(s).
- Page 2, lines 59 to 66, the last paragraph needs to mention what the author intends to do in this research, relating to the research aim and questions.
Materials and Methods
- Page 2, lines 69 to 94, there are some repetitive materials that have already been mentioned in the introduction section.
- Page 3, line 95, please add a north direction and scale bar in Figure 1.
- Page 4, line 106, Table 1 should be placed on top of the table.
- Page 4, line 107, Section 2.2 looks like a supplementary customer reservation price system in Qatar, rather than data analysis framework. Please either revise or clarify it.
- Page 5, lines 155 to 159, more detailed information is needed.
- Page 7, lines 216 to 217, some of the arguments and classifications need evidence. For instance, how do the 3 degrees of excludability being classified? This needs to be clarified.
- Page 7, line 221, some of the sentences need to be clarified. For instance, the sentence ‘the aim of this investigation is constrained to Qatar’ does not make sense.
- Page 7, line 229, Is Table 4 completed?
Outputs
- Page 8, lines 256 to 268, the results need to be elaborated. For instance, mean values are different. So what is the story behind the modelling?
- Page 8, lines 271 to 277, references are needed when discussing the correlation results.
- Page 11, lines 320 to 326, there are some typos here.
Discussion
- Page 12, lines 363 and 364, typos again.
Conclusions
- The author needs to summarise the paper, point out the key arguments/contributions, as well as providing policy implications.
Author Response
- Abstract: importance of customer reservation price included, sentence clarified, qualitative method application manner included
- Introduction: research background, aim questions, and research gaps clarified. Research goal mentioned.
- Materials and methods: Figure 1 scaled and north direction depicted. Table 1 placed on top. Section 2.2 clarified. Detailed secondary information provided. Classification of the 3 degrees of excludability clarified. Sentence clarification done. Table 4 completed.
- Outputs: Story behind the modelling depicted. References of the correlation results pin pointed. Typos amended.
- Discussion: typos amended.
- Conclusions: paper summarised, key arguments pointed out, and policy implications depicted
Round 2
Reviewer 2 Report
Dear author
Congratulations! I can see the improvement in your research.
Author Response
Thanks for improving my article.
Reviewer 3 Report
Dear authors, I could see the improvement in the paper. Although I would still prefer this research to be a better placed in the Introduction and Discussion into to current debate within the field, I accept the changes made on the manuscript.
Author Response
Thanks for improving my article
Reviewer 4 Report
I can see that the author has made some amendments. However, can the author please address each of my queries/comments point by point, as well as indicating the page and line numbers accordingly? Otherwise, I would not be able to see where the amendments were made. Thanks.
Author Response
Dear Sir/Madam,
I am writing to you in order to implement all your wise comments and amendments suggested by the reviewers.
All the modifications can be followed in the attached document.
I look forward to hearing from you soon.
Yours faithfully,
Jaime Larumbe
